# Modified Carboxymethylcellulose-Based Scaffolds as New Potential Ecofriendly Superplasticizers with a Retardant Effect for Mortar: From the Synthesis to the Application

**DOI:** 10.3390/ma14133569

**Published:** 2021-06-25

**Authors:** Clotilde Capacchione, Stephan Partschefeld, Andrea Osburg, Rocco Gliubizzi, Carmine Gaeta

**Affiliations:** 1Department of Chemistry and Biology “A. Zambelli”, University of Salerno, 84084 Fisciano, SA, Italy; cgaeta@unisa.it; 2BI-QEM SPECIALTIES S.p.A., 84021 Buccino, SA, Italy; rocco.gliubizzi@bi-qem.com; 3Department of Construction Chemistry and Polymer Materials, F.A. Finger Institute of Building Materials Science, Bauhaus-Universität Weimar, 99423 Weimar, Germany; andrea.osburg@uni-weimar.de

**Keywords:** superplasticizers, carboxymethylcellulose, mortar

## Abstract

This article is focused on the research and development of new cellulose ether derivatives as innovative superplasticizers for mortar systems. Several synthetic strategies have been pursued to obtain new compounds to study their properties on cementitious systems as new bio-based additives. The new water-soluble admixtures were synthesized using a complex carboxymethylcellulose-based backbone that was first hydrolyzed and then sulfo-ethylated in the presence of sodium vinyl sulphonate. Starting with a complex biopolymer that is widely known as a thickening agent was very challenging. Only by varying the hydrolysis times and temperatures of the reactions was achieved the aimed goal. The obtained derivatives showed different molecular weight (Mw) and anionic charges on their backbones. An improvement in shear stress and dynamic viscosity values of CEM II 42.5R cement was observed with the samples obtained with a longer time of higher temperature hydrolysis and sulfo-ethylation. Investigations into the chemical nature of the pore solution, calorimetric studies and adsorption experiments clearly showed the ability of carboxymethyl cellulose superplasticizer (CMC SP) to interact with cement grains and influence hydration processes within a 48-h time window, causing a delay in hydration reactions in the samples. The fluidity of the cementitious matrices was ascertained through slump test and preliminary studies of mechanical and flexural strength of the hardened mortar formulated with the new ecological additives yielded values in terms of mechanical properties. Finally, the computed tomography (CT) images completed the investigation of the pore network structure of hardened specimens, highlighting their promising structure porosity.

## 1. Introduction

Carboxymethylcellulose (CMC) is a renewable, biocompatible and versatile material obtained by modification of cellulose gained as a by-product of industrial processes in sugar firms [1]. This organic material is composed by modified glucopyranose units in which some hydrogen atoms of hydroxyl groups are replaced by carboxymethyl moieties with a different substitution degree. The glucopyranose monomers are linked by β-glycosidic bonds creating the final structure. This sugar-based material belongs to the family of derivatives of cellulose that are used in different business sectors, but due to its high versatility it represents one of the most interesting and employed backbones among them [2]. Figure 1 shows the chemical structure of carboxymethylcellulose.

Cellulose ethers have found an important role as water retention agents and regulators of rheological properties in building material applications as additives [4,5]. The field of application of such additives is quite diverse. Cellulose derivatives are increasingly used in plaster systems, tile adhesives and sprayed concretes, as they improve the tackiness and smoothness of binder in addition to water retention. Two theories exist on the mechanism of action of cellulose ethers in cement pastes. One is the adsorptive mechanism of action based on the fact that these high-molecular polymers adsorb by the anionic anchor groups (sulphonate, carboxylate and phosphate groups) on the positive charge sites of the binder surface. In this way, the cellulose derivatives reduce the pore diameter of the cement paste, which reduces the permeability. On the other hand, there is the idea of a non-adsorptive mechanism of action. The cellulose derivatives absorb a considerable amount of water molecules into their hydrate shell and physically bind the water. In addition, the hydrocolloids swell strongly due to the water absorption, which additionally clogs cement paste pores, which in turn contributes to the water retention capacity [6,7]. To this end, a forefront application of these materials is to modify and to employ them as additives in cement-based mortar, named superplasticizers, to provide more sustainable construction materials for the engineering field. Superplasticizer or SPs are intriguing organic compounds that have offered a vast potential of innovation in concrete technology since their discovery in the last century. Indeed, these additives are water-soluble compounds capable of dispersing the cement particles in a colloidal system, inhibiting their aggregation. This interesting microscopic interaction results into a macroscopic effect: a low-slump cementitious product can be easily transformed into a flowing, pourable, well workable material [8]. Cement is a construction material widely used to fulfil the societal demands and even more researchers are involved in studies to formulate innovative products with outstanding workability and physical properties [9]. SPs gave a huge contribution in the improvement in materials engineering during the 40 last years, starting from the first class of additives based on lignosulfonates [10,11]. Instead from the beginning of the 1970, the first synthetic products have been developed: naphthalene/melamine sulfonate derivatives and polycarboxylate esters, which represent the second and third generation of additives respectively [12]. The employment of these products as SPs has shown dangerous side effects that provide a growing concern for the ecological point of view. The human and environmental welfare are continuously threatened in several ways [13,14]. and Ruckstuhl et al. [15], in 2002, reported one example of the risk assessment related to the use of polynaphthalene sulfonate (SNF) on the aquifers proximal to construction sites. The case studied was focused on the investigation on the effect of their leached derivatives of the undergrounded waters quality. Some hazardous derivatives trickle down in the ground and they are condensed through biodegradation processes into oligomers that affect the healthiness of groundwater and the surrounding environment. For these reasons, researchers are drawn into driving innovation through the research of new, biocompatible and performing additives with a low environmental impact. It would represent a possible solution to preserve also non-renewable natural resources. It is highlighted from the literature that the use of ecological backbones such as modified carbohydrates will come to play a crucial role on the prospects of additives for cementitious systems [16,17]. The admixtures based on biodegradable starting materials can behave as agents with different properties on mortar and concrete systems. One example of putative carbohydrates as additives is given by the studies carried on by Shenghua Lv et al. [18]. They explored how modified cyclodextrins like poly-carboxymethyl-β-cyclodextrin superplasticizer, also named PCM-β-CD, behave as flow agents but with a strong effect of retardation as a function of the dosage of the additive. Cellulose ethers show a similar behavior on the hydration of Portland cement. In addition to the effect on the rheology and water retention of the cement, its hydration is sometimes greatly delayed. This is caused by the adsorption of the cellulose ethers on the clinker phases or first hydration products. Pourchez et al. [19,20] found in their investigations that both the hydration of the C_3_S and the hydration of C_3_A or the C_3_A-gypsum system are significantly changed. Lasheras-Zubiate et al. [21] studied one example of how chitosan derivatives bearing ionic and non-ionic groups could affect the properties of fresh cementitious systems. Due to its abundance in nature, it is considered second only to cellulose. Thanks to the introduction of polar functional groups, it was possible to increase its hydrophilic properties, an indisputably fundamental parameter for a superplasticizer. It was highlighted how chitosan derivatives can modify the working life of mortars and can behave as thickening agents. Therefore, the object of that study is to modify the features of a derivative of the cellulose. In particular, the aim of this work is to modify the features of a green backbone, the carboxymethylcellulose, that is a very well-known thickening agent, to study whether it was possible to modify and to use it as a flow agent in mortar. As it is highlighted from the previous studies, both molecular weight and the number of charges of additives are key parameters for the final performance of the additives. For instance, in 1988, Andersen et al. [22] carried on studies on SPs based on sulfonated polystyrenes with different molecular weight (Mw) and they observed that, considering the Mw as a function of the charge amount, there is a logarithmic trend of the reduction of the viscosity decreasing the Mw of the additive.

In addition, even more stress is placed on the resistance aspect related to hardened matrixes and recently, more literature papers reported how the strength development process [23] and durability [24] issues of cement mortar have been extensively studied.

According to the literature, it was performed the modification of the Mw and the introduction of anionic charges all along the chain of the CMC to carry on multiple investigations to understand their behavior on cementitious systems.

A combination of different analyses were employed such as the studies of the setting times, chemical investigation of the composition of the pore solution, calorimetric studies, dispersing abilities and trials on the hardened materials on an assessment of their possible application as SPs in the engineering field.

## 2. Materials, Procedures and Methods

### 2.1. Materials, Laboratory Apparatus and Methods

The CMC (≥ 99.5% of purity) and Sodium vinylsulfonate (25 wt. % solution in water) were purchased from Acros Organics™ (Fair Lawn, NJ, USA) and Alfa Aesar brand (Haverhill, MA, USA) respectively. The binder of this study is the Ordinary Portland Cement CEM II/A-LL 42.5 R (Opterra GmbH, Karsdorf, Germany) in accordance with the DIN EN 197-1. Dry solids content of SPs was determined to calculate the concentration of additives, weighing the dry residue until the total removal of the humidity contained in the samples. The water content of the fluidificants was considered and subtracted to the quantity required for making the cementitious pastes. X-ray diffractometer (Seifert XRD 3003 TT (New York, NY, USA) with Euler cradle and X-Y table and Rietveld refinement) were used for the X-ray analysis of the cement powder. The initial and the final setting point were ascertained through Vicat test measurements (Vicatronic MA-E044; TESTING; Berlin, Germany). The rheological properties of the cement slurries were investigated with the rotation viscometer (Rheotec^®^ Brookefield DV III-ultra, Middleboro, MA, USA) with SC-4 29 spindle (Middleboro, MA, USA). Differential and total heat release curves were obtained through isothermal calorimetric studies with the calorimeter (mc cal^®^ cement calorimeter, C3-Prozesstechnik, Gieboldehausen, Germany) at 20 °C for 72 h. The chemical composition of cement and the analysis of the pore solution composition of samples were carried on through the ICP-OES (inductively coupled plasma optical emission Spectroscope, Aktiva M, Horiba, Kyoto, Japan). Spectral photometer (Schott Instruments, UvLine 9100, SI Analytics, Mainz, Germany) was used for the adsorption experiments of the CMC SPs on the cement grains. Spread tests performed were compliant to the current legislation using the CEN Standard Sand following the DIN EN 196-1. Strengthening behavior was determined through the ultrasonic velocity analysis (Ultratest^®^ IP-8, Ultratest, Achim, Germany). Compressive and flexural tests followed respectively the ASTM C 349 and the ASTM C 348. A strength testing machine (TIRAtest 28100, Tira, Schalkau, Germany) was used. The grout samples were scanned with X-ray CT (phoenix nanotom M research edition; waygate technologies; Wunstorf, Germany) to investigate through 3D images the pore structure network.

### 2.2. Synthesis and Characterization of Starting Materials

#### 2.2.1. Carboxymethylcellulose Modification—Overview

The chemical modification of the CMC was performed in two steps. First, the CMC was reduced in molecular weight by acid hydrolysis, and in the second step, anionic charges were implemented into the molecular structure. The Figure 2 shows an overview of the synthesis workflow.

#### 2.2.2. Acid Hydrolysis Procedures at Variable Temperatures and Reaction Times

The acid hydrolysis of CMC was performed by revisiting the procedure previously reported in the literature that was used for the sulfo-ethylation process of the degraded scaffolds [22,25]. Twenty-five grams of CMC were poured into 200 mL of ethanol (Carl Roth GmbH & Co. KG, Karlsuhe, Germany) and stirred for 5 min. Then 10 mL of an aqueous solution of HCl (37%) (Carl Roth GmbH & Co. KG, Karlsruhe, Germany) was added to the mixture at room temperature, and then the mixture was heated at 70 °C and stirred for a variable time (24, 48 and 72 h). Then the reaction mixture cooled to room temperature and added of 100 mL of an aqueous solution of NaOH (Carl Roth GmbH & Co. KG, Karlsruhe, Germany; purity 98 wt.%, 10% *w*/*w*) under stirring. The solid phase was separated by centrifugation at 2215 G for 5 min (Eppendorf^®^ Centrifuge 5804 R; Hamburg, Germany) and washed with ethanol (3 × 100 mL). A pale-yellow solid was obtained that was stored at 30 °C for 72 h.

#### 2.2.3. Sulfo-Ethylation of the Hydrolyzed Substrates

The sulfo-ethylation of the hydrolyzed products was performed starting from 5 g of the hydrolyzed backbone. The hydrolyzed products were mixed at 30 °C for 12 h in 200 mL of deionized water. Then, the reaction temperature was set at 70 °C and added of 1.2 g of NaOH (Carl Roth GmbH & Co. KG, Karlsruhe, Germany; purity 98 wt.%) and stirred for 60 min. Then, 16.2 g of sodium vinyl sulfonate (25% aqueous solution) (ThermoFisher GmbH, Kandel, Germany) was added dropwise. The reaction mixture was stirred at 70 °C for different times (see Table 1), then cooled and stored at room temperature (RT).

### 2.3. Cementitious Paste Analysis

#### 2.3.1. Initial and Final Setting Time Evaluation

The determination of the point of the beginning and the end of the setting of mortar was defined preparing cementitious paste of the only reference setting the water/cement ratio(w/c) at 0.5 and starting with 450 g of cement powder and 225 g of deionized water. The automatic penetration of the needle was carried with Vicat apparatus in a defined time lapse, after a delay of 2 h to occur the setting points in accordance with the ASTM C191 specifications (Standard Test Method for Time of Setting of Hydraulic Cement by Vicat). The reference points of the beginning and of the end of the loss of plasticity were defined respectively after 326 and 444 min from the mixing of the grout.

#### 2.3.2. Rheological Properties Investigations

The thixotropic properties of the cement slurries were so studied preparing grout pastes following these conditions: 30 g cement, 15 g of deionized water at w/c = 0.5 with 1% by weight of cement (bwoc) percentage of additive. Immediately after the mixing, the specimen was immediately insert into the sample loader with the spindle. The test was performed for each sample three times and a mean value was calculated. The results per each SP were compared with the reference without additive. Each experiment was performed three times and a mean value was determined. Each specimen required additionally six measured values at each rotation speed to determine the flow and viscosity curves to finally understand the rheological behavior of the cementitious paste.

#### 2.3.3. Isothermal Calorimetric Studies of Early-Stage Hydration Processes

The hydration process was studied through calorimetry at 20 °C. Per each sample, there were followed the same conditions: w/c = 0.5 and 0.5% of additive. The 5 g of cement powder were added to the solution of water and SP. After mixing the slurry in the vial, it is then inserted inside the calorimeter as previously described in the literature [26,27]. The heat flow was measured for 48 h. Each calorimetric measurement was performed three times and a mean value of the heat curves was reported.

#### 2.3.4. Investigations of the Composition of the Pore Solution Analysis

Different stages of the hydration process of the cement were investigated on triplicate samples through the analysis of the composition of the liquid phase of the suspension made with lime solution, cement powder and additives. At the ratio between lime and cement of 20 and 0.5% of active, 5 g of cement were mixed with the 0.02 M Ca(OH)_2_ water solution and with 0.5% bwoc of SPs for several defined times: 15 min, 1 h, 4 h, 6 h, 12 h, 24 h and 48 h. The hydration reaction was stopped mixing the centrifuge tubes at 2215 G at 20 °C, the supernatant was filtered, and 1 mL was diluted 20 mL of HNO_3_ to stabilize the pore solution for the ICP-measurements. The error rate during the ICP-measurement depends on the content of the respective ion type in the pore solution. For ions with a content less than 50 mg/mL, the error is 1%, while the error for ions with a content of more than 50 mg/mL is 2%.

#### 2.3.5. Adsorption Studies of CMC SPs on Cement Grains through UV-Vis-Spectrophotometry

The investigations were carried out following the phenol method already reported by used in the literature at a wavelength of 490 nm [28]. An aqueous solution of 0.02 M Ca(OH)_2_ (Carl Roth GmbH & Co. KG, Karlsruhe, Germany; purity 96 wt.%) was prepared to mimic the alkaline conditions of the prepared mortar at a w/c ratio of 0.5. Each sample was prepared with a ratio between the water solution and cement of 0.5 and 0.5% of additive by weight of cement. The suspension was stirred for 2 h at room temperature. After the mixing time, the supernatant was separated from the precipitate by centrifugation at 2215 G for 20 min and the supernatant was filtrated (membrane filters with a 0.45 µm pore size) and used for SPs detection. 5 wt% phenol solution (Carl Roth GmbH & Co. KG, Karlsruhe, Germany) and 96 wt% sulfuric acid (Carl Roth GmbH & Co. KG, Karlsruhe, Germany) were added to the supernatant. This exothermic reaction develops a yellow color solution. The concentration of the blank and the sample containing the additive is measured with a spectral photometer at a wavelength of 490 nm. The adsorption studies were carried on kinetically and the average of 10 absorbance measurements for each sample is recorded.

The concentration of the superplasticizers in the supernatant was determined by using a calibration line with various concentrated glucose solutions and comparing the concentration values obtained with those of the blank without additive.

#### 2.3.6. Computed Tomography Analysis

Grout specimens were prepared in cylindrical vials (radius: 5 mm, height: 30 mm) starting from 5 g of anhydrous cement, deionized water, w/c = 0.5 for the reference and w/c = 0.5 and 1% bwoc of additive. The vials were sealed with a lid and stored at RT. All the samples were scanned in triplicate after 7 and 28 days.

### 2.4. Mortar Investigations

#### 2.4.1. Workability Investigations through Slump Test

The samples for the workability tests were prepared mixing cement, deionized water, sand and superplasticizers. Mortar mixer was used for mixing inert materials, cement, deionized water and SPs into a cementitious paste following these conditions: 0.5% bwoc; w/c = 0.5; CEM II 42.5 R = 450 g (DIN EN 197-1); H_2_O = 225 g; Standard sand = 1350 g (DIN EN 196). After the mix, to evaluate the trend of the additive on mortar, the spread values were obtained measuring in a specific time lapse the mean values of three measurements of diameters after 4′, 30, 60, 90 and 120 min. The sand for the Slump tests, setting time estimation and for the mechanical tests is the CEN-standard sand in accordance with the current legislation ASTM C143 (Standard Test Method for Slump Hydraulic-Cement Concrete). The investigations were carried on with the water/cement ratio w/c = 0.5, using 0.5% and 1% bwoc of the additives.

#### 2.4.2. Strengthening Behavior by Means of Ultrasonic Investigations

For the ultrasonic velocity measurements, the obtaining of the cementitious paste required 300 g of cement, 150 mL of deionized water for the only CEM II 42.5 R. The investigation of the setting times of the samples was performed using the 0.5% bwoc of each additive repeating all the experiment three times. From of the only reference, the CEM II 42.5R it was estimated that the ultrasonic velocity was 1417 m/s for the initial and 1794 m/s for the final point. By the comparison of the results of the ultrasonic velocity of the samples with the one of the references, the characterization of the values was extrapolated per each sample.

#### 2.4.3. Estimation of the Mechanical Properties of the Hardened Mortar

For the bending and compressive strength tests, the specimens were prepared following the method DIN EN 197-1 and the specimens were cured until the test ages of 7 and 28 days into prism shapes (40 mm × 40 mm × 160 mm) at 85% rel. humidity. The prism shaped samples were prepared complying to the legislation in force and three measurements were determined for each specimen. Average values were obtained per each investigation. Bending stress was applied on the long surface of prism specimens using a bend tester (ASTM C 348) even on triplicate samples.

## 3. Results

### 3.1. Chemical and Mineralogical Composition of CEM II/A-LL 42,5R

The chemical and mineralogical compositions of CEM II 42.5 R are reported in Table 2 and Table 3.

### 3.2. Studies of the Retardant Effect of Synthetised SPs on Cement with Setting Times Determination

The investigated setting time points of the only CEM II 42.5 R and of the synthetized additives with the w/c = 0.5 and a dosage of 0.5%. The initial and the final setting time of the only reference were respectively 326 and 444 min.

### 3.3. Rheology Investigations

The rheologic properties of the grouts doped with the synthetized SPs were assessed by means of rheometer as shown in the Figure 3 and Figure 4. The values of shear stress and dynamic viscosity of the cement CEM II 42.5R as a function of the rotation speed were measured and the shear rate of all the synthetized products. From the investigation of the rheological properties of the starting materials and of the sample S1 and S2, that are biodegradable backbones hydrolyzed at room temperature and sulfo-ethylated for 24 h and 48 h at 70 °C respectively, it was observed that the only CEM II 42.5 R shows typical profiles as highlighted in the literature and the additives S1 and S2 behave as thickening agents, so the measurements were not recorded [16,29,30,31]. The materials S3–S7 (Table 1) hydrolyzed at 70 °C, showed the capability to reduce gradually both the shear stress that the dynamic viscosity of the cement CEM II 42.5R that as reported in the Figure 4, regarding the sample S5 in blue, indeed these values were very close the value of the only cement reference.

From this empirical observation, it is clear how the rheological parameters are strongly related with the molecular weight of the material and with the ionic groups anchored on the scaffolds that can influence both the shear stress and dynamic viscosity values [32]. An improvement of these parameters were observed also for the samples S4–S7 obtained after prolonged hydrolysis for 48–72 h. The sample S5 showed values very close to the reference (blu line in Figure 4 and Figure 5), while S6, obtained by longer times of hydrolysis and sulfo-ethylation (72 h), showed a lower shear stress and dynamic viscosity values (green line in Figure 4 and Figure 5).

### 3.4. Calorimetry

By calorimetric study it is possible to investigate how classify additives from the investigation of their effect on the different phases that have been identified in the hydration process of the cement [33]. The graph in Figure 5 illustrates the influence of the cellulose-based superplasticizer (w/c ratio 0.5, by addition of 0.5% bwoc of additives) on the kinetics of the hydration of CEM II/A-LL 42.5 R in early stage by isothermal calorimetry. That stage is very sensitive to the adding of the additives indeed the only reference displays a dormant period of 2.5 h. The rate of the hydration and the heat release curves showed a different length of the dormant period when carbohydrate-based SPs were added to the pastes, as already studied in the literature. In particular, the analysis of the most significant results obtained from calorimetric studies showed that cellulose extended the quiescent period because it initially chelated the first mineralogical phase involved, the aluminates. Calcium sulphate normally forms ettringite by chelating aluminates that interact less quickly with water [34,35]. CMC SPs amplify this effect by stabilizing the ettringite layers, causing a delay in the initial hydration of the aluminates and consequently slowing down the quiescent phase. A strong retardant effect was observed on the quiescent period before the hydration of the C_3_S by employing the following samples: the maximum extension of the dormant period was observed with the raw CMC and the samples S3 (hydrolyzed and sulfo-ethylated for 24 h) that both showed a very similar trend. Prolonging the time of hydrolysis until 48 h, with the sample S4. A smaller quiescent period was observed that was still too long at almost 6 h. Finally, with the samples S5, S6, S7 we observed a mild retardant effect of almost 4 h of the hydration of the main clinker phases involved into this step. Reducing the molecular weight of the additives, the distention of the dormant period was reduced. Moreover, observing the heat release of the samples, this value was very low in the sample S3 and with the raw CMC, that highlights the strong retardant effect of these samples. The samples S4–S7 were obtained by raising the temperature of hydrolysis (Table 1) and consequently showed a lower Mw; interestingly for these samples the maximum main heat release value was observed compared to the reference cement. For example, the maximum of thermal energy released with the only reference is 14.5 J/g·h a value significatively higher that measured for the sample S7 of 4.28 J/g·h. After 48 h, the analysis of the heat release curves shows that the samples reach the same heat released values as the single reference at the same degree of hydration. This means that the SP CMCs influence the hydration process only for 48 h, after this period the samples reach the same degree of hydration like the reference.

### 3.5. Composition of the Liquid Phase of the Pores Network

The analysis of the main ions contained into the pore solution was carried on during outgoing process of hydration of the main clinker phases [36,37]. Different stages were investigated through ICP measurements of the solutions, and it was confirmed the retardant effect of the CMC-based SPs already observed with the previous analysis. The most abundant ionics species found in the liquid phase were Ca^2+^, Si^4+^, Al^3+^, S^6+^, K^+^ and Na^+^, followed by small traces of Mg^2+^ and Fe^3+^ [38]. Each sample doped with the synthesized SPs, showed the same behavior, for these reasons it was reported only the analysis of the sample S6, where single ion abundance was compared in a specimen obtained with only the cement and with the adding of the SP as reported in Figure 6. The increase of the Ca^2+^ content in the sample doped with the additive was seen over a longer time. The Al^3+^ concentration reflects the production of the hydrate products obtained by the fast aluminate reactions that occur during the initial stage of the cement hydration. The levels of the Al^3+^ in the first hour of reaction were lower than the corresponding level in the reference, highlighting the retardant effect typical of carbohydrates-based additives. Coherently, even the values of the level of S^6+^ and the Si^4+^ decrease slowly during time. K^+^/Na^+^ concentrations follow the same trend in the sample doped with S6 and in the only reference. After the dormant period, the second hydration process occurs, and it involved the silicates. The reactions led the formation of portlandite and Ca(OH)_2_ that increased the pH value of the already alkaline environment of cement. By focusing on the graph in Figure 7, it is possible to observe how in the only reference, the pH fluctuated around 12 and after 6 h, it reached the maximum value that is shifted around 12 h from the mixing of the cement slurry with the sample S6.

### 3.6. Adsorptive Behavior of Admixture on Cement Particles Surface

Cement grains can adsorb negatively charged superplasticizer and this interaction is facilitated by their mosaic surface: indeed, it is composed by different mineral phases which counter-ions drive the adsorption of SPs molecules on cement grains. When the mineral phases of cement come into contact with water, a heterogeneous surface charge distribution is observed, which it is fundamental for the electrostatic force. CMC SPs were not detected into the supernatant phase, which meant the occurrence of the complete binding of the additives on the mineral surfaces through the electrostatic interaction between cement surface and the −SO_3_ groups added through the sulfo-ethylation step. The adsorption on cement particles is facilitated both by the −SO_3_ groups and by the structure of the polymer, which with its anionic carboxylate groups can chelate Ca^2+^ on the cement surface. This leads to the formation of a layer of SPs on the surface that cannot come into contact with water, slowing down the hydration process and altering the growth and morphology of the hydration products [39].

### 3.7. Porosity of Cement Pastes

Cement matrix is characterized by relatively high porosity. The structural network of cement is composed of two types of pores of different dimensions: gel pores (d < 10 nm) and capillary pores (pores size/diameter d > 10 nm or larger capillary pores size 50 nm < d < 10 μm can also be observed if there are air voids into the matrix) [38]. To study this aspect, computed tomography defined the percentage of porosity of the hardened materials after 7 and 28 days. The pore network structures of samples S5, S6 and S7 were compared with the single reference as shown in Table 4. The computed porosities showed how at higher number of charges and at lower Mw, the porosity of the final materials improves, which means a higher durability against environmental influences. The reduction of air bubbles and capillary pores were observed which are very important for the durability of the cement matrix. These SPs increased the durability of the hardened matrix over time which will not be affected by time and external factors such as environmental pollution. External agents, such as water, can enter the network, creating cracks and damaging the structure. The reduction of bubbles and capillary pores using SPs is an important factor to improve the strength and the durability of the cementitious material and these aspects were investigated by CT measurements. Through CT measurements the contribution of CMC-SPs on the pores, their diameter and size were investigated. The results clearly show the reduction of air bubbles and of the size of the capillary pores. After 7 days, the main porosity decreases in a range of 40 to 61% in comparison to the reference. The Sample with S5 shows the highest decrease in porosity after 28 days of 82%. In addition, the CT images in Figure 8 show a significant reduction of air voids.

### 3.8. Applicative Tests: Slump Tests

The investigation of dispersing performance of CMC-based SPs was carried out by slump tests on mortar. The studies of the effect of the samples were conducted at w/c = 0.5 and with a percentage of additive of 1% bwoc. The results were compared with the mortar obtained without the SPs as a reference that does not display any fluidizing property, which are shown in Figure 9. Samples S1, S2, S3 (Table 1) on the mortars behave as thickening agents and cannot fluidize the cement system. As illustrated in the Figure 9, the reference alone does not show any fluidizing properties, while the mortar doped with CMC-SPs showed that at 0.5% bwoc it activates good fluidity. In fact, sample S4, hydrolyzed at high temperature for 48 h, starts to behave as a fluidizing agent but only with samples S5, S6 and S7 were excellent values of spreading diameters and setting times measured. In fact, the evolution of the fluidity of the paste doped with S5, S6 and S7 was measured over time and all the samples showed an initial spreading diameter of ca. 240 mm remaining almost constant for 2 h. This shows that the synthesized additives can extend the workability time within the first 120 min when the doped fresh mortars are very flowable.

### 3.9. Strengthening Behavior by Means of Ultrasonic Investigations

The starting and final points of setting obtained with the Vicat test (see Section 3.4) correspond to an ultrasonic velocity of 1417 and 1794 m/s, which is shown in Figure 10. The length of the delay periods of the synthesized products are then investigated: the only raw CMC never sets after 5 days, the samples S2, S3 and S4 generated a serious retardation effect until more than 36 h. Reducing the Mw of the samples at high temperature, with the samples S5, S6 and S7 the initial and final setting times were shorter than the cement-only sample, which is considered as reference, and the previous modified samples still behaved as retardant agents: indeed the values of the initial and final setting time obtained with the sample S5 were T_i_ = 798 min and T_f_ = 912 min, respectively. A slightly reduced setting point was defined with the specimen S6; the beginning and the end of the hardening was after 609 and 729 min. Finally, the delay period of the sample S7 started after 930 min and ended after 1069 min.

### 3.10. Properties of the Hardened Mortar

Studies on the mechanical behavior of hardened mortar under compressive and flexural stresses after 28 days showed a trend: increasing the anionic charge state and reducing the molecular weight, the mechanical properties are slightly increased from 78 to 93 N/mm^3^ regard to the compression parameter of hardened cement mixed with bio-based superplasticizers but this is still not so relevant if compared with the reference. Moreover, the flexural parameter remains a relatively vulnerable aspect as reported in Figure 11. For each SP, three mortar prisms were prepared, and the measurements were recorded in triplicate and the mean values were reported.

## 4. Conclusions

This article focused on the research and development of new cellulose ether derivatives. Several synthetic strategies have been pursued to obtain new compounds for the study of their fluidizing properties as new super fluidizers for mortar systems. These backbones are based on water-soluble carboxymethylcellulose scaffolds that have shown an interesting plasticizing effect on mortar. Carboxymethylcellulose is a biodegradable and complex biopolymer that is widely known for its behavior as a thickening agent. Indeed, lots of examples in the literature mentioned its role as an increase in the viscosity of the cement. Moreover, the effect of thickening also determines the increase of the compactivity of the hardened cement paste. Indeed, during the hydration reactions of cement calcium-silicate hydrate (CSH) and calcium hydroxide (CH) are produced in more quantities, but it was demonstrated by literature that the CMC can modify the volume fractions of CSH and CH, favoring the growth of the CSH phase that determines the hardening of final matrices. With these studies, it was interesting to add new properties to the CMC, the flowability. It was really intriguing to understand the duplicable nature of this green compound: CMC showed to be able to switch from a thickening agent to a flowable one, reducing its molecular weight and increasing the availability of anionic charges all over its backbone in an easy procedure.

Therefore, the values of shear stress and dynamic viscosity were strictly dependent on the Mw weight because only rising the temperature of hydrolysis from the room temperature to 70 °C it was possible to reduce the dimension of the backbones. Moreover, only through the combination of the acid hydrolysis and the subsequent addition of anionic fillers to the main backbone, it was easy to modify its physical–chemical properties. In fact, the reduction of its Mw and the possibility of the charges to interact with the cement grains made these additives potential candidates as new superplasticizers, confirming the data previously reported in the literature. Through workability tests it was observed that the spread increased by addition of the modified CMC SPs from 155 up to 240 mm. Furthermore, the setting time as well the calorimetric experiments showed a retarding effect on cement hydration by all modified CMC SPs. In particular, the dormant period was significantly prolonged for several hours. This behavior was also confirmed by the analysis of the pore solution. It was found that the concentration of calcium, aluminum and silicon increase much lower than the reference without additives. This effect was caused by high adsorption rate of the CMC SPs on the hydrating cement particles, which significantly delays the hydration reaction, as shown by the adsorption experiments. The preliminary studies on hardened mortar, through mechanical investigations and CT analysis of the mortar, have highlighted the promising properties of these new bio-based admixtures that are able to reduce the porosity of the final hardened material, for instance with the most promising sample S7, the porousness after 7 days varied from the 40% of the only reference to 61%, up to 82% at the end of the curing time of 28 days and coherently less air voids were observed through the CT investigations.

These new derivatives broaden the spectrum of possible ecological superplasticizers to propose an environmentally friendly alternative to the family of petrochemical additives currently on the market, and finally provide new insights into the engineering of building materials. It is not excluded the possibility that in the future, both the flowable effect of the synthetized CMC SPs and with their behavior as retarding admixtures will be studied on different binders such as gypsum to amplify the window of workability of this building material that generally does not show high flowability and has a fast setting time.

Further studies such as the determination of the anionic charge of the synthetized SPs can highlight the way they interact with the cementitious surface. Moreover, investigations of their effect on hardened matrices in the long term in different climatic conditions could look beyond the properties of the pastes doped with these bio-based compounds and foresee their potential employment in the yards.

## Figures and Tables

**Figure 1 materials-14-03569-f001:**
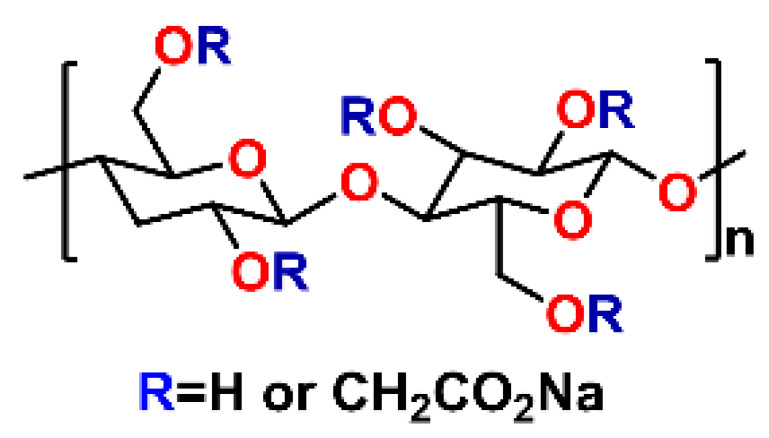
Chemical structure of carboxymethylcellulose [3].

**Figure 2 materials-14-03569-f002:**
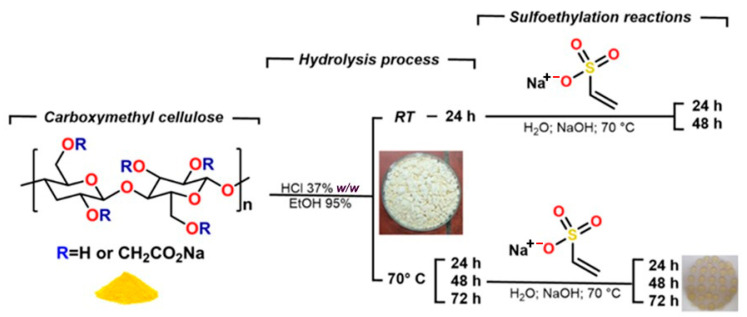
Overview of the synthesis workflow.

**Figure 3 materials-14-03569-f003:**
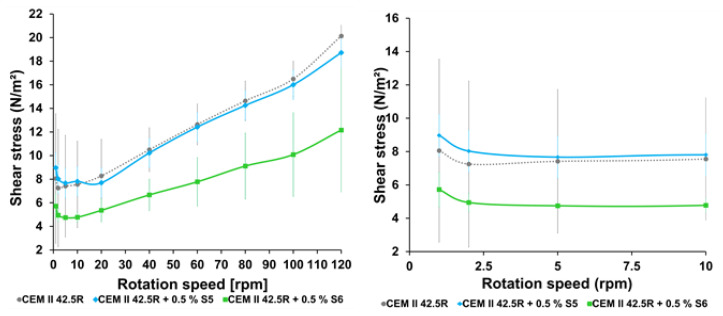
Investigation of the shear stress in dependence on rotation speed ((**left**) full graph, (**right**) enlargement)) of the cement CEM II 42.5R at w/c = 0.5 with 0.5% of additives. Test conditions: 0.5% bwoc of SPs; w/c = 0.5; CEM II 42.5R = 30 g (EN197-1); deionized H_2_O = 15 g.

**Figure 4 materials-14-03569-f004:**
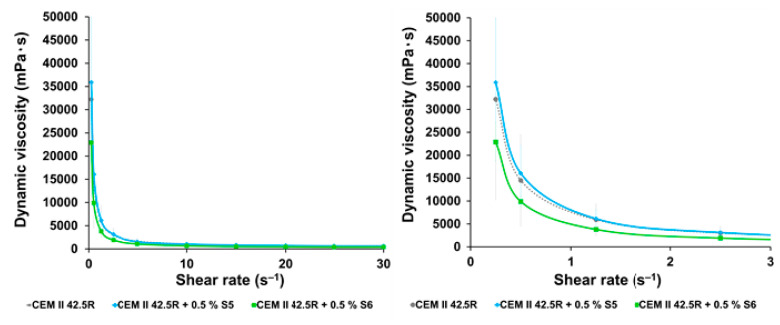
Investigation of dynamic viscosity (**left**) full graph, (**right**) enlargement) of the cement CEM II 42.5R at w/c = 0.5 with 0.5% of additives. Test conditions: 0.5% bwoc of SPs; w/c = 0.5; CEMII 42. 5R = 30 g (EN197-1); deionized H_2_O = 15 g.

**Figure 5 materials-14-03569-f005:**
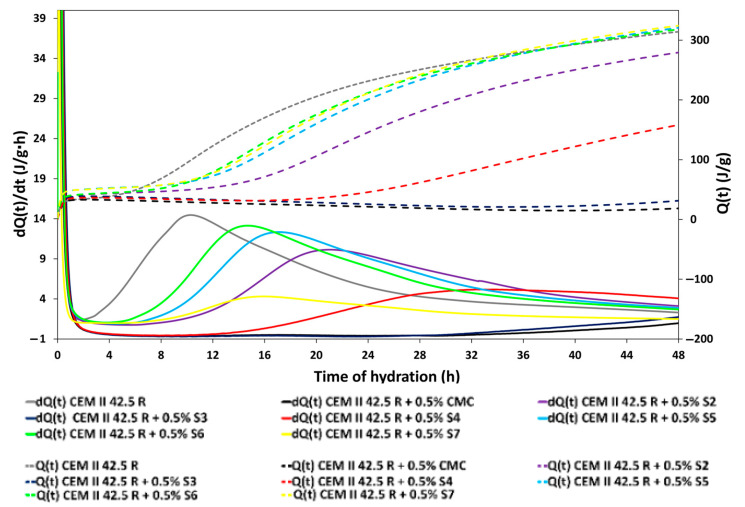
Influence of the cellulose-based superplasticizer investigated with a w/c ratio = 0.5 on the early age hydration of CEM II 42.5 R by calorimetric studies.

**Figure 6 materials-14-03569-f006:**
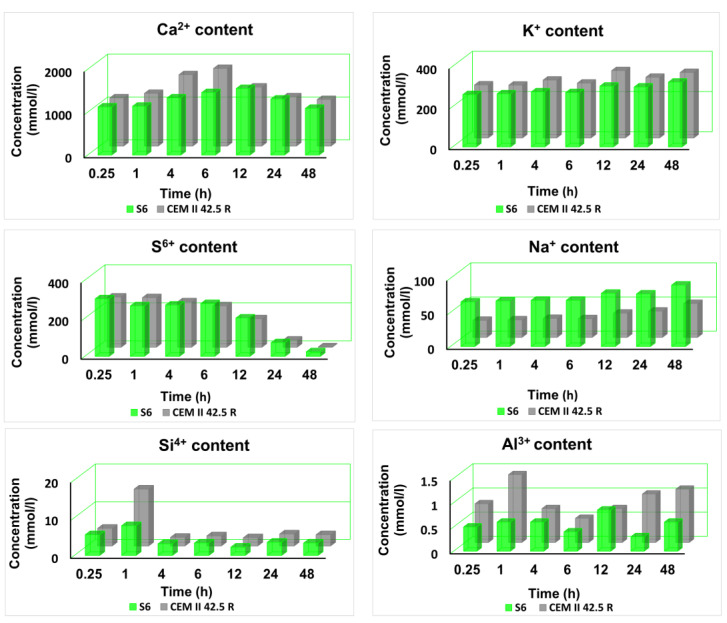
Pore solution chemistry of the only CEM II 42.5 R (grey) and of CEM II 42.5 doped with S6 (Table 1) (green).

**Figure 7 materials-14-03569-f007:**
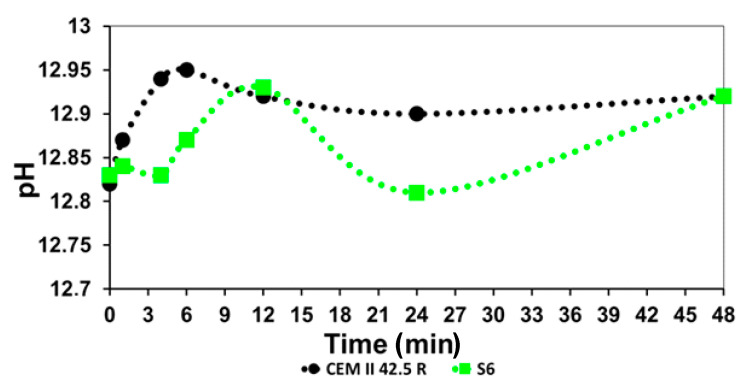
Evolution of the pH value during the outgoing process of hydration of cement (grey) and of the cement doped with the sample S6 (green) (Table 1).

**Figure 8 materials-14-03569-f008:**
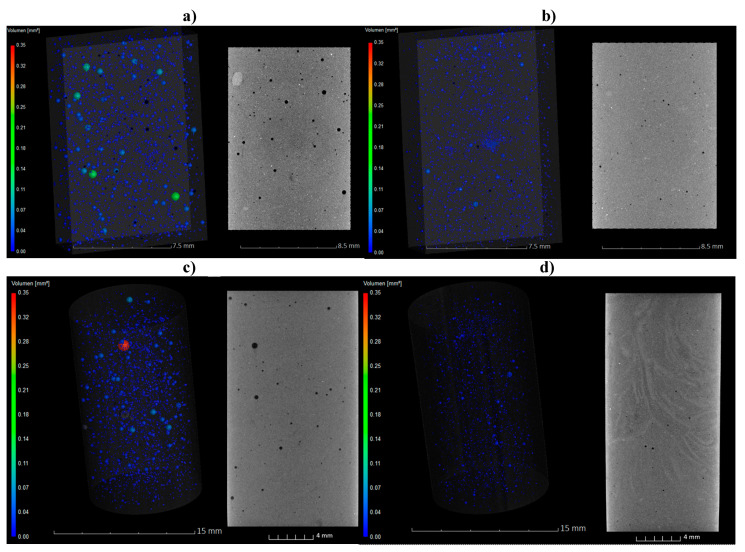
3D and 2D area void fractions of the only reference CEM II 42.5 R (**a**) after 7 and (**b**) after 28 days compared with the speciments of S7 (Table 3) (**c**) after 7 and (**d**) after 28 days.

**Figure 9 materials-14-03569-f009:**
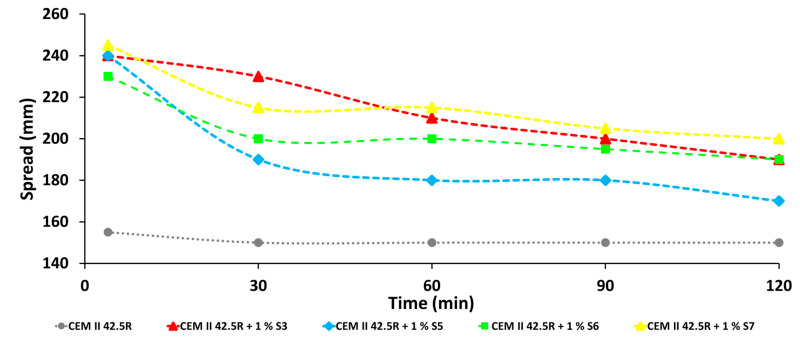
Evaluation of the fluidification properties through slump test of CEM II 42.5 R doped with and 1% bwoc of additives with a w/c = 0.5 (DIN EN 197-1).

**Figure 10 materials-14-03569-f010:**
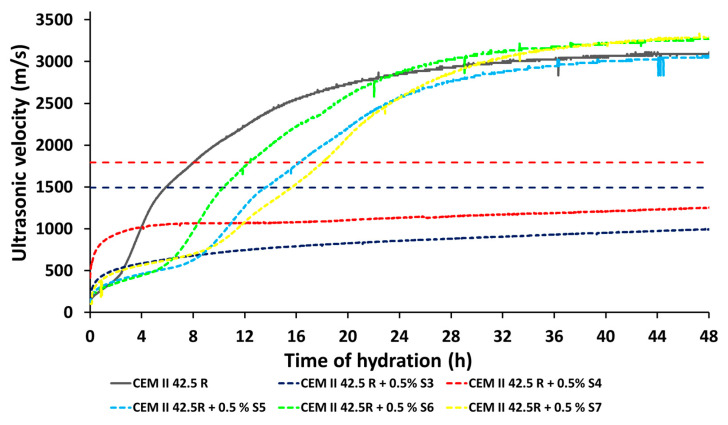
Study of the setting time through Ultrasonic measurements of the CEM II 42.5 R at w/c = 0.5 and with 0.5% bwoc of additives.

**Figure 11 materials-14-03569-f011:**
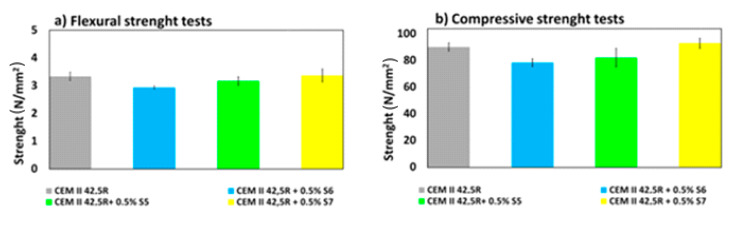
Mechanical properties for (**a**) flexural strength and (**b**) compressive strength of the mortar, as the only reference and the mortar doped with S5 (blue), S6 (green) and S7 (yellow) (Table 1) and at w/c = 0.35 and 1% bwoc after 28 days.

**Table 1 materials-14-03569-t001:** Library of synthesized materials at different temperature and times of hydrolysis and sulfo-ethylation.

Specimens	Temperature of Hydrolysis (°C)	Time of Hydrolysis (h)	Time of Sulfo-Ethylation at 70 °C (h)
S1	RT *	24	24
S2	RT *	24	48
S3	70	24	24
S4	70	48	24
S5	70	48	48
S6	70	72	72
S7	70	72	72

* RT: room temperature.

**Table 2 materials-14-03569-t002:** Chemical Composition of CEM II/A-LL 42.5R.

Chemical Composition of CEM II/A-LL 42.5R (% wt)			
SiO_2_	18.4	K_2_O	1.1
Al_2_O_3_	5.4	Na_2_O	0.2
Fe_2_O_3_	2.8	SO_3_	3.3
CaO	62.0	Other Oxides	1.0
MgO	1.4	Annealing Loss *	5.2
TiO_2_	0.19	Drying Loss **	0.2
MnO	0.03	-	-

* Annealing Loss at 950 °C; ** Drying Loss at 105 °C.

**Table 3 materials-14-03569-t003:** Mineralogical composition of the CEM II/A-LL42,5R by X-Ray Diffraction and Rietveldt.

Mineral Phase	(% wt)
C_3_S	55 ± 1.4
C_2_S	8.6 ± 1.1
C_3_A	6.8 ± 1.2
C_4_AF	8.3 ± 1.0
Calcite	10.5 ± 0.8
Arcanite	2.3 ± 0.6
Gypsum	2.0 ± 0.7
Bassanite	1.5 ± 0.8
Anhydrite	3.0 ± 0.6
Portlandite	0.7 ± 0.5

**Table 4 materials-14-03569-t004:** X-ray computed tomography to analyze the porosity of mortar samples after 7 and 28 days.

Specimens	Porosity after 7 Days (%)	Porosity after 28 Days (%)
CEM II 42.5 R	46	33
S5	25	6
S6	38	12
S7	28	9

## Data Availability

All the data is available within the manuscript.

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
