# Peer review of "Modified Carboxymethylcellulose-Based Scaffolds as New Potential Ecofriendly Superplasticizers with a Retardant Effect for Mortar: From the Synthesis to the Application"

_materials, 2021, doi:10.3390/ma14133569_

Round 1
Reviewer 1 Report
Comments
This paper studied the ecofriendly superplasticizers for mortar. The outcome is interesting for readers. However, there are several aspects that need to be improved. The reviewer can only recommend for publication if the author satisfactorily address the following major comments in the revised version.
- The author need to mention the standard test methods/document (e.g., ASTM) number used in this study.
- How many replicate samples were tested in each category?
- The novelty of the study should be highlighted more clearly at the end of introduction section. How this study is different from the published study in literature?
- How the outcome of this study will benefit researchers and end users? This need to be highlighted in introduction or end of conclusion.
- The strength development process of mortar should be discussed in introduction section to improve the background study. Recently, the strength development process [Ref: Characteristics, strength development and microstructure of cement mortar containing oil-contaminated sand] and durability [Ref: Ageing of particulate-filled epoxy resin under hygrothermal conditions] issues of cement mortar are studied. Suggest to include them in introduction section with proper citations to improve the background study.
I would be happy to see the revised version to understand how these comments are being addressed.
Reviewer 2 Report
The authors obtained a large amount of experimental data on the properties of the superplasticizer based on carboxymethylcellulose. In general, the presented work corresponds to the theme of the journal. The reviewer has a number of questions and comments.
1. The introduction, in my opinion, should be made more capacious. It is necessary to strengthen the introductory part, give more references to works on their objects of study. In my opinion, too much of the description of the environmental risks of materials based on naphthalene / melamine sulfonate derivatives and polycarboxylate esters, NFS, SNF in the literature review.
2. There are typos on the test, for example, lines 98 and 341 (the ion charge is incorrect).
3. What is the reason for the choice of the temperature of acid hydrolysis of CMC (70°C)?
4. Some of the methods and instrument names described in section 2.1. (2.1 Materials, laboratory apparatus and methods) are duplicated again below. Does this make sense? For example, the instruments in XRF and ICP-OES in section 2.1. and section 2.2.4; 2.1. and 2.3.5 (spectral photometer
5. What are the errors in determining Ca2+, Si4+, Al3+, S6+,K+, Na+ by ICP-OES (composition of the pore solution analysis)?
6. Most units have the country of manufacture and city listed, some do not. Uniformity should be adhered to.
7. If the specific surface area of the studied components (CMC, products of its modification, cement samples) was determined, these data should be given.
8. Conclusions should be specified, specify numerical values of the most important obtained results. How economically feasible to use such additives on an industrial scale?
Reviewer 3 Report
Congratulations for your case study on superplasticizers for mortar based on cellulose ether derivatives.
I consider that your work addresses na interesting topic that is suitable for Materials journal.
In general, this paper is clear, the topic is contextualized, the description of the methodology used in laboratory work is very detailed and all the references are pertinent. However, I have found some minor mistakes that need to be correct:
Line 233 - Please replace EN197-1 by DIN EN197-1 and EN196 by DIN EN196
Line 248 - Please replace 2.4.2. Estimation of the … by 2.4.3. Estimation of the …
Line 254 - Please insert 4. Before Conclusions
Line 400 – Please replace Table 4 by Table 3
Also, Figure 3 is missing. So, please check all the document in order to renumbering figures.
In relation to the experimental work, I think the parameters considered were appropriated, the experiments were well conducted according to standards and both data collection and analysis are well conducted. However, in my opinion, a specimen for control should be added.
Round 2
Reviewer 1 Report
I have no further comments
Reviewer 2 Report
Dear Authors! Thank you for your comprehensive answers!